# Comparison of Nuclear Medicine Therapeutics Targeting PSMA among Alpha-Emitting Nuclides

**DOI:** 10.3390/ijms25020933

**Published:** 2024-01-11

**Authors:** Kazuko Kaneda-Nakashima, Yoshifumi Shirakami, Yuichiro Kadonaga, Tadashi Watabe, Kazuhiro Ooe, Xiaojie Yin, Hiromitsu Haba, Kenji Shirasaki, Hidetoshi Kikunaga, Kazuaki Tsukada, Atsushi Toyoshima, Jens Cardinale, Frederik L. Giesel, Koichi Fukase

**Affiliations:** 1Laboratory of Radiation Biological Chemistry, FRC, Graduate School of Science, Osaka University, Toyonaka 560-0043, Japan; 2MS-CORE, FRC, Graduate School of Science, Osaka University, Toyonaka 560-0043, Japan; yoshifumi_shirakami@irs.osaka-u.ac.jp (Y.S.); kadonaga.yuuichirou.med@osaka-u.ac.jp (Y.K.); watabe.tadashi.med@osaka-u.ac.jp (T.W.); ooe@rirc.osaka-u.ac.jp (K.O.); toyo@irs.osaka-u.ac.jp (A.T.); koichi@chem.sci.osaka-u.ac.jp (K.F.); 3Department of Science, Institute for Radiation Sciences, Osaka University, Suita 565-0871, Japan; 4Nuclear Medicine, Graduate School of Medicine, Osaka University, Suita 565-0871, Japan; 5Radioisotope Research Center, Institute for Radiation Sciences, Osaka University, Suita 565-0871, Japan; 6Nishina Center for Accelerator-Based Science Nuclear Chemistry Group, RIKEN, Wako 351-0198, Japan; xiaojie.yin@riken.jp (X.Y.); haba@riken.jp (H.H.); 7Laboratory of Alpha-Ray Emitters, Institute for Materials Research, Tohoku University, Sendai 980-8577, Japan; kshira@imr.tohoku.ac.jp; 8Research Center for Electron Photon Science, Tohoku University, Sendai 982-0826, Japan; kikunaga@lns.tohoku.ac.jp; 9Research Group of Heavy Element Nuclear Science, Advanced Science Research Center, Japan Atomic Energy Agency, Naka-gun 319-1195, Japan; tsukada.kazuaki@jaea.go.jp; 10Nuclear Medicine Department, University Hospital Düsseldorf, 40225 Düsseldorf, Germany; jens.cardinale@med.uni-duesseldorf.de (J.C.); frederik.giesel@med.uni-duesseldorf.de (F.L.G.); 11Natural Product Chemistry, Graduate School of Science, Osaka University, Toyonaka 560-0043, Japan

**Keywords:** astatine-211, actinium-225, PSMA, cytotoxicity, double strand brake, reproductive capability

## Abstract

Currently, targeted alpha therapy (TAT) is a new therapy involving the administration of a therapeutic drug that combines a substance of α-emitting nuclides that kill cancer cells and a drug that selectively accumulates in cancer cells. It is known to be effective against cancers that are difficult to treat with existing methods, such as cancer cells that are widely spread throughout the whole body, and there are high expectations for its early clinical implementation. The nuclides for TAT, including ^149^Tb, ^211^At, ^212/213^Bi, ^212^Pb (for ^212^Bi), ^223^Ra, ^225^Ac, ^226/227^Th, and ^230^U, are known. However, some nuclides encounter problems with labeling methods and lack sufficient preclinical and clinical data. We labeled the compounds targeting prostate specific membrane antigen (PSMA) with ^211^At and ^225^Ac. PSMA is a molecule that has attracted attention as a theranostic target for prostate cancer, and several targeted radioligands have already shown therapeutic effects in patients. The results showed that ^211^At, which has a much shorter half-life, is no less cytotoxic than ^225^Ac. In ^211^At labeling, our group has also developed an original method (*Shirakami Reaction*). We have succeeded in obtaining a highly purified labeled product in a short timeframe using this method.

## 1. Introduction

There are many nuclides that could be used in targeted α therapy (TAT) (Table 1) [1]. Among them, astatine-211 (^211^At) and actinium-225 (^225^Ac) are thought to be useful α-emitting nuclides (Table 2). This is because these nuclides can be produced in relatively large quantities [2]. Prostate specific membrane antigen (PSMA) is highly expressed in metastatic and castration-resistant prostate cancers and is a well-known therapeutic target for prostate cancer in radioligand therapy [3,4,5]. Although the functions of PSMA in cancer cells are still unclear, its expression increases in correlation with the degree of cancer progression, making it an extremely useful theranostic target for prostate cancer. Prostate cancer is mainly seen in people over 60 years old, and it is known that more than half of men over 80 years old have latent prostate cancer [6]. Remarkable results in TAT have been reported using ^225^Ac-labeled therapeutics, and their usefulness is clear [7]. However, the supply of ^225^Ac is insufficient for widespread clinical application. The current supply of ^225^Ac is mostly due to the decay of uranium-223 (^233^U), with a worldwide supply of approximately 63 GBq (2 Ci)/year [8]. Therefore, lutetium-177 (^177^Lu), a β-ray emitting nuclide, is used and [^177^Lu]PSMA-617 has already been approved and is commercially available in the US and Europe [9]. Initial treatment for prostate cancer generally includes surgery or radiation therapy, followed by hormone therapy. Recently, various treatments have been implemented for hormone-sensitive prostate cancer (HSPC). Castration-resistant prostate cancer (CRPC) has a poor prognosis, and taxane chemotherapy, such as docetaxel (DTX) and cabazitaxel (CBZ), is administered, but the treatment effect may not be sufficient. In the TheraP trial, it was reported that ^177^Lu-PSMA-617 treatment had fewer side effects and lowered PSA levels compared to CBZ [10]. We are currently conducting research with ^211^At, which can be produced using an accelerator, and have successfully labeled astatine as a highly selective PSMA compound [11]. We previously showed the strong therapeutic effect of this compound and its promising potential for clinical applications.

This study aimed to clarify the performance of the PSMA-targeting compound labeled with ^211^At (^211^At-PSMA-5) in comparison to ^225^Ac-PSMA-617. These nuclides differ in their physical half-lives (^225^Ac; 10 days, ^211^At; 7.2 h), the number of α-particle emissions (^225^Ac; 4, ^211^At; 1), and in the properties of the elements themselves (^225^Ac; actinoid, ^211^At; halogen), resulting in the need for different binding domains for labeling. Even when using the same α-particle emitting nuclide, it is thought that there are other problems of practical application other than the amount of supply. Examples include the ease of labeling, the amount of compound used, and efficacy. If a small amount of the compound is used, the side effects caused by the compound might be reduced to a minimum. However, no comparison has been made to date between α-emitting nuclear medicine therapeutics that have the same molecular targets, e.g., PSMA. We have conducted this experiment in the hope that it will be useful for selecting the optimal nuclide for each situation.

Nuclear medicine involves minimally invasive procedures. However, by fully understanding the principles and properties of nuclear medicine therapeutics, we believe that they are effective. Furthermore, through an investigation of the dynamics and pathological analysis of nuclear medicine, it has been shown that the side effects are not significantly different from those of other drugs [12,13,14,15]. Nuclear medicine therapeutics are expected to become new treatment options for patients for whom existing therapeutic drugs are not suitable. To expand patient options, we hope to demonstrate scientific evidence of its effectiveness, especially in TAT.

**Table 2 ijms-25-00933-t002:** Comparison of physical properties between ^211^At and ^225^Ac.

	^211^At	^225^Ac
**Half-life**	7.2 h	10 days
**Maximum** **α energy (MeV)** **and emission rate**	5.87–41.8%	5.83–100.0%
**Tissue range**	55 to 70 μm	47 to 85 μm
**Particle**	α (2 routes)	4α, 2β
**Effective dose rate constant** **(μSv·m^2^·MBq^−1^·h^−1^)**	0.0058	0.0027
**Source**	[16]	[17]

## 2. Results

### 2.1. Evaluation of Effects on Cell Viability

We seeded the cell line LNCaP, which is known for its high PSMA expression, and the cell line PC3, characterized by low PSMA expression, at a density of 1 × 10^4^ cells/well in a 96-well plate. The cells were treated with ^225^Ac-PSMA-617 or ^211^At-PSMA-5 for 3 days, and cell viability was evaluated. The toxicity of the labeled PSMA was more pronounced in LNCaP cells than in PC3 cells (Figure 1). The ^225^Ac-PSMA-617 nuclide had a stronger effect on cell viability than ^211^At-PSMA-5; however, above a certain concentration, cell viability did not decrease in a dose-dependent manner.

### 2.2. Evaluation of Effect on Replication

We performed a colony formation assay to assess the extent to which ^225^Ac-PSMA-617 and ^211^At-PSMA-5 affected the replication ability of the cells. Figure 2 shows a numerical graph based on the photographs shown in Figure 3. Cytotoxicity occurred in both PC3 (PSMA^low^) and LNCaP (PSMA^high^) cells in a concentration-dependent manner (Figure 2A and Figure 3A). This effect appeared to be stronger in cells with high PSMA expression. When the cells were treated with ^225^Ac-PSMA-617, dose-dependent inhibition of replication was observed in both PC3 and LNCaP cells. After treatment with ^211^At-PSMA-5, no effect was observed in PC3 cells (Figure 2B and Figure 3B).

### 2.3. Evaluation of Cytotoxicity

DNA double strand breaks (DSBs) were observed in both PC3 (PSMA^low^) and LNCaP (PSMA^high^) cells (Figure 4). Figure 5 depicts a numerical graph based on the photographs shown in Figure 4. Green fluorescence indicates the presence of γH2AX, a marker of DSB. Focies of γH2AX were strongly induced by ^225^Ac-PSMA-617 (Figure 5A), and many focies were induced by ^211^At-PSMA-5 (Figure 5B). When comparing their effects on (a) PC3 and (b) LNCaP cells, more foci were found in the LNCaP cells, indicating that many DSBs appeared in the LNCaP cells.

### 2.4. Uptake of ^225^Ac-PSMA-617 or ^211^At-PSMA-5

The ^225^Ac-PSMA-617 nuclide had a low background of intracellular uptake and was hardly taken up by PC3 (PSMA^low^) cells (Figure 6a). In contrast, ^211^At-PSMA-5 was taken up in a certain amount by PC3 cells and large amounts of uptake were observed in LNCaP (PSMA^high^) cells (Figure 6b).

### 2.5. Inhibition of Unlabeled Chemicals

The inhibition of uptake by unlabeled chemicals using LNCaP cells (PSMA^high^) is shown in Figure 7. When we confirmed the presence of competitive inhibition by unlabeled chemicals for uptake at the same dose, it was clear that ^211^At-PSMA-5 (Figure 7b) was inhibited at a lower concentration than ^225^Ac-PSMA-617 (Figure 7a). The IC_50_ of ^225^Ac-PSMA-617 was 2.64 nM and that of ^211^At-PSMA-5 was 0.32 nM. Non-specific binding was evaluated using PC3 cells (PSMA^low^). The effects of non-labeled chemicals on both the uptake of ^225^Ac-PSMA-617 (Figure 7c) and ^211^At-PSMA-5 (Figure 7d) in PC3 cells were thought to be minimal.

### 2.6. Stability of ^225^Ac-PSMA-617 and ^211^At-PSMA-5

The stability of ^211^At-PSMA5 was evaluated using HPLC, TLC, and electrophoresis. Its stability in blood and urine, along with the absence of metabolites and degradation products, was detected in vitro experiments. In in vivo experiments, slight (<1%) deastatination was observed. The same evaluations were conducted on ^225^Ac-PSMA-617, confirming that it was stable.

## 3. Discussion

In this study, ^225^Ac-PSMA-617 and ^211^At-PSMA-5 were compared in terms of their nuclide decay number (Bq) in order to focus on the performance of the labeled chemicals. This study revealed that ^225^Ac, which contains many α-particles, is highly cytotoxic, as was expected (Figure 1). The strong cytotoxic effect of ^225^Ac-PSMA-617 was reflected in the loss of replication ability (Figure 2 and Figure 3). At first glance, ^225^Ac-PSMA-617 seemed to have stronger cytotoxicity than ^211^At-PSMA-5. However, considering the AUC and α-ray emission ratio, it was difficult to say that ^225^Ac-PSMA-617 was more effective than ^211^At-PSMA-5 because there was a 25 to 83-fold difference even with the same radioactivity (Bq). We also compared γH2AX as an indicator of radiation-induced DNA damage (DNA double strand breaks, DSBs), and it became clear that ^211^At-PSMA-5 induces DNA damage at the same levels of ^225^Ac-PSMA-617 (Figure 4 and Figure 5). This was thought to be because ^211^At-PSMA-5 was taken up by cells in larger amounts and acted closer to the nucleus (Figure 6). Labeled chemicals appear to work more effectively as nuclear medicine therapeutics for intracellular uptake. It is interesting to note that more ^211^At-PSMA-5 was taken up by the cells even though both chemical structures are similar (Figure 8 and Figure 9). However, considering their structures, there seems to be a slight difference in fat solubility. Since ^211^At-PSMA-5 is slightly more lipophilic, it is presumed to have a higher affinity for the cell membrane (Figure 7). The correction for non-specific binding concerning IC_50_ should be strictly calculated using the inhibition value in an experimental system using PC3 cells (PSMA^low^), Xenopus oocytes, or HEK293 cells overexpressing PSMA. However, for both ^225^Ac-PSMA-617 and ^211^At-PSMA-5, the ratio of PC3 to LNCaP cell uptake at 120 min did not change (approximately 30%) (Figure 6). Even when the background was subtracted at the same rate, the relationship between the IC_50_ of ^225^Ac-PSMA-617 and ^211^At-PSMA-5 was similar. Thus, the IC_50_ was calculated from the LNCaP value (Figure 7). The IC_50_ value of ^225^Ac-PSMA-617 was approximately eight-fold higher than that of ^211^At. The lipid solubility of chemicals is an important factor in nuclear medicine. This is because a certain degree of fat solubility improves tumors [18,19,20,21]. However, if fat solubility is too high, adsorption onto plastic experimental tools will be high. For example, an increase in the amount of adsorption on the purification column results in a poor collection yield. Additionally, when administered to animals, it has been observed that excretion from the intestinal tract increases and the amount excreted in feces increases. Screening for suitable compounds should consider both accumulation and excretion.

^225^Ac is an excellent therapeutic-emitting nuclide, but its current supply is limited. PSMA is also highly specific and only has a few side effects (caused by damage to the salivary and lacrimal glands, such as dry eye [22] and xerostomia [23]). Non-specific accumulation over a long period would induce side effects. If the ratio of non-specific accumulation is the same, the nuclide with a shorter half-time is less likely to induce side effects due to non-specific accumulation. The ^211^At-PSMA-5 nuclide requires significantly fewer chemicals for labeling than ^225^Ac-PSMA-617, and its cost of manufacturing is also lower. In this study, we attempted to compare two α-emitting nuclides used in TAT. We believe that our results suggest that, even though its half-life is much shorter than ^225^Ac, ^211^At can be sufficiently tolerated in clinical use by effectively utilizing its properties. If the half-life is short, the hospitalization period for nuclide decay can be shortened. This might reduce the burden on patients.

Although there are many clinical reports of ^225^Ac-PSMA-617 use in humans, there are only a few reports on animal experiments using ^225^Ac. Additionally, these reports were published later than the human reports. The ^225^Ac-PSMA-617 nuclide demonstrated high levels of accumulation in the liver. However, it is well known that ^225^Ac alone accumulates in the liver. There are doubts as to whether the results showing accumulation only in the liver can depict metastasis [24]. A previous study used a combination of the PD-L1 antibody and ^225^Ac-PSMA-617 in RM1-PGLS mice. Although it is used in combination with an immune checkpoint inhibitor, it has not been as effective as expected [25,26]. Rodent metabolism is different from that of primates, meaning that research results on rodents are not guaranteed to be applicable to humans. In contrast, ^211^At-PSMA-5 was investigated for imaging in cynomolgus monkeys by Watabe et al. Additionally, no acute inflammation was observed, and the side effects were expected to be minimal. Although clinical trials using ^211^At-PSMA-5 have not yet been conducted, animal studies have shown positive results [11]. Thus, we expect that future clinical trials of ^211^At-PSMA-5 will yield good results.

Irradiation to the target with high linear energy transfer (LET) radiation is highly lethal. Thus, resistance to this radiation is unlikely to occur. Therefore, selecting a specific and excellent molecular target, such as PSMA, is essential for nuclear medicine. To achieve a stable supply of nuclides, selecting optimal molecular targets, establishing chemicals that recognize and label these targets, and collaborating across departments, such as medicine, science, and nuclear physics, is necessary.

We also confirmed the utility of ^225^Ac in another molecular target, fibroblast activate protein α (FAPα) [27,28]. However, as previously mentioned, its supply is insufficient for clinical use [8]. Various research groups are attempting to find methods to produce it in large quantities; however, no satisfactory method has yet been established. Therefore, there is an urgent need to establish a method for producing ^225^Ac. Currently, we believe that the most promising method in Japan is the use of an electron linear accelerator. The production of ^225^Ac was conducted using ^226^Ra and a linear accelerator at Tohoku university [^226^Ra(γ, n)^225^Ra→^225^Ac]. In Japan, the National Institute for Quantum Science and Technology (QST) and Nihon Medi-physics, Co. Ltd. are also trying to produce it with the transmutation of ^226^Ra to obtain ^225^Ac [^226^Ra(p, 2n)^225^Ac] [29]. In Canada’s particle accelerator center (TRIUMF), a method of obtaining ^225^Ac through the nuclear spallation of thorium-232 [^232^Th(p, spall)^225^Ra→^225^Ac] is being attempted [30]. In contrast, ^211^At has a short half-life; therefore, it must be manufactured in large quantities. This must be performed efficiently because it uses an accelerator. It is desirable to create multiple bases within a country where supplies are needed. However, to transport them to places where transportation is difficult, it is necessary to create large quantities. In 2023, Haba et al. aimed to develop targeted irradiation technology that minimizes the loss of ^211^At due to radioactive decay and increases the production efficiency of ^211^At through high-intensity beam irradiation. Their results may be useful for the mass production and supply of ^211^At.

The half-life of ^211^At is 7.2 h and α decays to bismuth-207 (^207^Bi with a half-life of 32 years) with a probability of 41.8%. Therefore, with a probability of 58.2%, it can become polonium-211 (^211^Po) through electron capture decay. Since ^211^Po undergoes α decay immediately (with a half-life of 0.52 s) to the stable isotope lead-207 (^207^Pb), ^211^At emits α-particles with virtually 100% probability. On the other hand, there are many daughter nuclides of ^225^Ac, such as francium-221 (^221^Fr, with a half-life of 0.12 μ seconds), ^217^At (with a half-life of 20 milliseconds), ^213^Bi (with a half-life of 47 min), ^213^Po (with a half-life of 3.65 μ seconds), thallium-209 (^209^Tl, with a half-life of 1 h), ^209^Pb (with a half-life of 3.3 h), and ^209^Bi (stable), all of which have short lifetimes. Because none of these generate radon as daughter nuclide, which is a gas, post-administration management is easy. For practical use, it is very important that the nuclides are easily handled. In addition to the two nuclides discussed in this research, there are several other α-emitting nuclides that may be used for TAT, as shown in Table 1. However, various choices should be made with consideration of the balance between demand and supply.

We are currently developing nuclear medicine therapeutics using the ^211^At and ^225^Ac nuclides. During this period, we have observed the advantages and disadvantages of each nuclide. Initially, we thought that ^225^Ac-PSMA-617 would provide better results than ^211^At-PSMA-5 and we hoped that ^211^At-PSMA-5 would be the next choice. However, the performance of ^211^At-PSMA-5 was unexpectedly good. High-power accelerators are currently needed to obtain ^211^At. However, since it can be obtained using accelerators, it is easier to obtain than ^225^Ac, particularly in Japan. Although the availability of nuclides differs from country to country, it can be said that, at least in Japan, astatine has been shown to be extremely useful as a labeled nuclide for nuclear medicine therapeutics. We are currently conducting an investigator-initiated clinical trial of astatine-labeled chemicals (Na^211^At, targeting thyroid cancer) at Osaka University Hospital. By demonstrating the usefulness of nuclear medicine therapeutics, especially α-emitting nuclear medicine therapeutics, we hope that this will become a new choice for patients who are currently unable to undergo surgery or for whom existing drugs are not effective.

The most important step in the development of nuclear medicine is the selection of molecular targets. If expressed in normal tissues as well as in cancer tissues, nuclear medicine therapeutics can accumulate in normal tissues and damage them, causing side effects. Therefore, it is desirable to use molecular targets with a higher specificity. In the development of compounds, even for the same molecular target, selectivity can vary significantly depending on the structure of the compound. In addition, side effects can occur when the labeled nuclide is released from the compound. This is because ^211^At behaves in a similar fashion to iodine and ^225^Ac accumulates in the bones and liver [31]. Consideration is also required in order to ensure safety. Key considerations in the clinical application of ^225^Ac-PSMA-617 include liver uptake, which is assumed to result from the decay products of ^225^Ac-PSMA-617. DTPA is an abbreviation for diethylenetriamine penta-acetic acid, a chelate compound with affinity for metals, and is a type of chemical protective agent against radiation damage. For example, Techne^®^ DTPA Kits (PDRadiopharma, Inc., Tokyo, Japan) or Indium (^111^In) DTPA Injections (Nihon Medi-Physics Co., Ltd., Tokyo, Japan) exist. This protective agent has the function of removing radioactive materials from the body and is said to be most effective in excreting radioisotopes. DTPA may be added to the formulation of ^225^Ac-PSMA-617 to prevent uptake in the liver (faster renal clearance). In the case of ^211^At-PSMA-5, the properties of the element are different from ^225^Ac; therefore, it may be necessary to use a different drug.

The functions of PSMA itself in cancer tissues are gradually being elucidated. For example, research conducted by Watanabe, et al. revealed the existence of PSMA-positive tumor endothelial cells in human prostate tumors, which enhance tumor angiogenesis in prostate cancer tissues [32]. The elucidation of the role of PSMA in cancer tissues has supported its importance as a molecular target. On the other hand, reports on the structure and dynamics of chemicals are also being considered [33]. We are also conducting studies, but because the in vivo environment and the stability of compounds are interrelated, the results are often not what we expected. We hope that similar studies conducted by various groups will clarify this relationship.

## 4. Materials and Methods

The ^211^At nuclide was acquired from RIKEN through a supply platform for short-lived radioisotopes. The ^211^At nuclide was produced according to the ^209^Bi(α, 2n)^211^At reaction and was separated from the Bi target using the dry distillation method. The separated ^211^At was then dissolved in pure water [11]. The ^225^Ac nuclide was produced mainly by members of Tohoku University [34,35]. Experiments after isolation were conducted at Osaka University. The intensity of the nuclides were measured using a germanium semiconductor detector (BE2020, Canberra, Mirion Technologies, Inc., Atlanta, GA. USA) and a γ-counter (Wizard^2^ 2480, PerkinElmer, Inc., Shelton, CT, USA). The samples treated with ^225^Ac were maintained until radiative equilibrium was reached before the measurements were taken, according to a previous study [31].

### 4.1. Structure and Preparation of ^225^Ac-PSMA-617 and ^211^At-PSMA-5

The PSMA-selected chemicals were labeled with each nuclide using previously reported procedures. These structures are shown in Figure 8 and Figure 9. The PSMA-5 precursor was synthesized at the Peptide Institute. Inc. (Osaka, Japan) for the *Shirakami Reaction*. The labeling method is explained in detail in the next section. To evaluate the quality of the labeled chemicals, we used a previously reported method.

### 4.2. Nuclide Production and Chemical Labeling

#### 4.2.1. Production and PSMA-5 Labeling of ^211^At

PSMA-5 was used as a highly selective PSMA compound to label with ^211^At. This compound has been previously reported by Watabe et al. We determined this compound to be optimal based on the labeling efficiency and in vitro and in vivo experimental results [11]. The method we used for labeling PSMA-5 was the “borono group-astatine exchange reaction”, also known as the “Shirakami reaction” [36].

#### 4.2.2. Production and PSMA-617 Labeling of ^225^Ac

The labeling method for ^225^Ac was established according to a previous study [37]. The ^225^Ac used for labeling was separated at the Tohoku University Institute for Materials Research and was then transported to Osaka University for use. PSMA-617 was dissolved in DMSO (1 mg/mL) and a 10% DMSO aqueous solution was prepared. This solution was mixed with 0.2 M AcONH_4_ and 10% Ascorbic acid and was incubated in 80 °C for 2 h. After measuring the dose using a Curie meter (ICG-8; ALOKA, Ltd., Tokyo, Japan), the quality was confirmed using electrophoresis and it was then used in experiments.

### 4.3. Cell Culture

The PC3 and LNCaP cells were obtained from RIKEN and ATCC cell banks, respectively. The cells were maintained in RPMI1640 (Fujifilm Wako Pure Chemical, Osaka, Japan) supplemented with 10% heat-inactivated fetal bovine serum (Thermo Fisher Scientific, Waltham, MA, USA) and 1% penicillin-streptomycin (Fujifilm Wako Pure Chemical). Sodium pyruvate (Fujifilm Wako Pure Chemical) was added to the LNCaP culture medium at a concentration of 1%. The cells were maintained using trypsin-EDTA (Fujifilm Wako Pure Chemical), according to standard methods. Both cell lines were in the logarithmic growth phase at the time of experimental preparation.

### 4.4. Evaluation of Cell Viability

Two days before treatment, the cells were seeded in 1 × 10^4^ cells/mL in 96-well culture plates. The cell numbers were measured using TC-20^TM^ (Bio-Rad Laboratories, Inc., Hercules, CA, USA). After 1 h of treatment, the cells were cultured for three days. Cell viability was evaluated using a cell counting kit-8 (Dojin, Kumamoto, Japan), according to the manufacturer’s protocol. The absorbance was measured at 450 nm using a MultiSkan FC (Thermo Fisher Scientific).

### 4.5. Colony Formation Assay

Both LNCaP (PSMA^high^) and PC3 (PSMA^low^) cells were seeded in 24-well plates and treated with labeled PSMAs at various concentrations. After 1 h of treatment, the cells were peeled and seeded at 1000 cells per well. Colony formation was observed for two weeks. After observation, the cells were fixed with 1% crystal violet solution. Colony formation was calculated by analyzing the coverage ratio of cells based on photographs using ImageJ software (https://imagej.net/downloads, accessed on 1 April 2023) [38].

### 4.6. Evaluation of DNA Double Strand Breaks

The cells were seeded in a culture chamber (WATSON, Tokyo, Japan). After treatment with labeled chemicals, the cells were fixed and did the immunofluorescence staining, according to a previously reported protocol [37]. The cells were observed using a BZ-X810 microscope (KEYENCE, Osaka, Japan). The fluorescence intensities were analyzed using ImageJ software [38].

### 4.7. Uptake of ^225^Ac-PSMA-617 and ^211^At-PSMA-5

Both LNCaP (PSMA^high^) and PC3 (PSMA^low^) cells were collected after 5, 30, 60, and 120 min of treatment, washed with PBS (-) three times, lysed using 0.1 N NaOH solutions, and collected into microtubes. The sample counts were measured using a γ-counter and their counts were corrected for the amounts of cell protein, according to a previous published protocol [39]. We also measured the protein amounts in the cell suspension using a protein assay BCA kit (Fujifilm Wako Pure Chemical), according to the manufacture’s protocol. The absorbance was measured at 570 nm using a MultiSkan FC.

### 4.8. Inhibition Assay

Based on previous research [40], the LNCaP cells were simultaneously treated with unlabeled chemicals (PSMA-617 or PSMA-5). For the inhibition assay, the cell uptake experiments and procedures were the same, except for treatment with the compound as an inhibitor. The amount of unlabeled compound added was more than 100 times that of the labeled compound.

## 5. Conclusions

Let us consider the respective absorbed doses of ^211^At-PSMA-5 and ^225^Ac-PSMA617 in the cell experiments. Since the cell treatment conditions were the same, it can be assumed that the masses of cells were the same. In this case, the energy value and the absorbed dose was proportional. In other words, it is thought that comparisons can be made in becquerel, considering the half-life. For example, assuming that there is no washout from the cells, considering the area under curve (AUC) for 3 days, ^225^Ac-PSMA-617 is about 6.3 times more powerful than ^211^At-PSMA-5. Considering that ^225^Ac emits four α rays, it is about 25 times more powerful. However, the amount of uptake itself is halved (Figure 6), which is approximately 12 times as much. There is no more than a 10-fold difference in their effects on the cells (Figure 4). Therefore, it can be said that PSMA-5 is superior as a compound. It might be better to consider the stability of PSMA-5 in more detail, but we consider that this stability might be increased by inducing three unnatural amino acids (Figure 9). PSMA-5 is expected to be sufficient for clinical use.

It might be said that ^211^At-PSMA-5 is more easily taken up by LNCaP cells than ^225^Ac-PSMA-617, suggesting that PSMA-5 might be superior as a compound (Figure 6 and Figure 7). On the other hand, whether it is better in clinical practice may depend on the balance between effectiveness and potential side effects, aside from the availability of the nuclide. Although more studies may be needed, our findings indicate the possibility of completing treatment regimens faster because of the shorter half-life of ^211^At-PSMA-5 for repeated administration compared to ^225^Ac-PSMA-617.

## 6. Patents

Patent No. JP7237366B2, JP7232527B2.

## Figures and Tables

**Figure 1 ijms-25-00933-f001:**
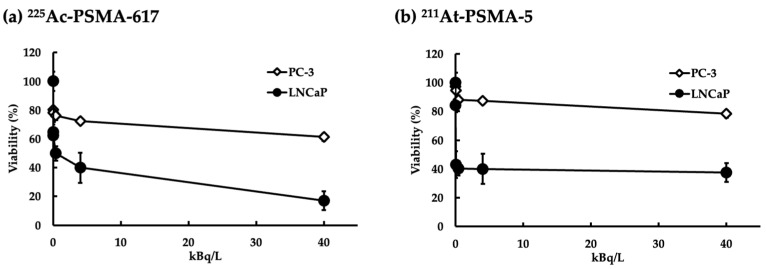
Cell viability following PSMA-targeted radioligand administration in LNCaP (PSMA^high^) and PC3 (PSMA^low^) cell lines. Diamonds represent PC3 cell lines and circles represent LNCaP cell lines. The mean ± S.E of the triplicate results is shown.

**Figure 2 ijms-25-00933-f002:**
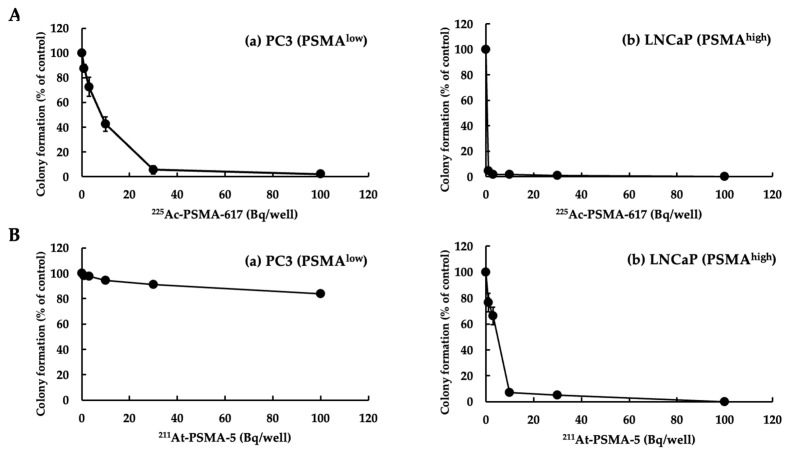
Colony formation (% of control). (**A**) (**a**) PC3 (PSMA^low^) and (**b**) LNCaP (PSMA^high^) cells treated with ^225^Ac-PSMA-617. (**B**) (**a**) PC3 and (**b**) LNCaP cells treated with ^211^At-PSMA-5. The mean ± S.E of the triplicate results is shown.

**Figure 3 ijms-25-00933-f003:**
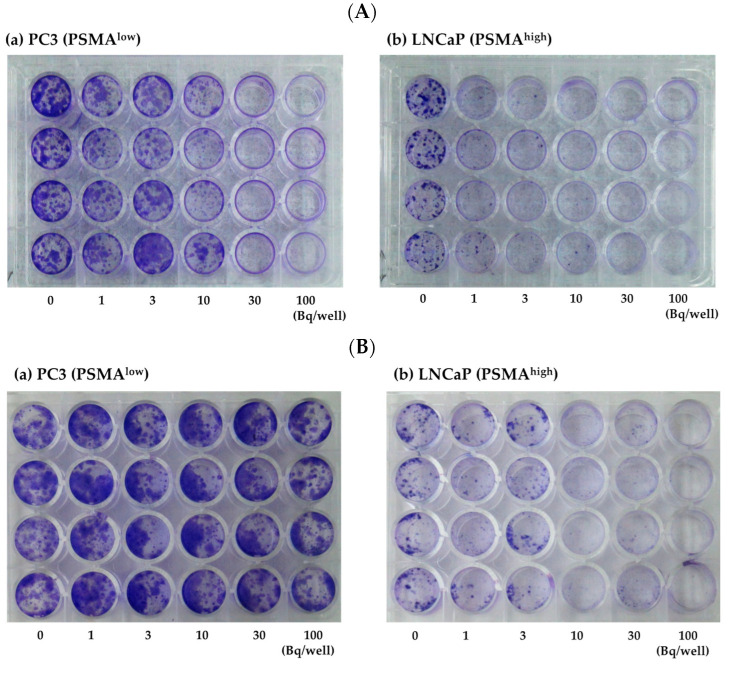
Cell images. (**A**) (**a**) PC3 (PSMA^low^) and (**b**) LNCaP (PSMA^high^) cells treated with ^225^Ac-PSMA-617 and stained with crystal violet. (**B**) (**a**) PC3 and (**b**) LNCaP cells treated with ^211^At-PSMA-5 and stained with crystal violet.

**Figure 4 ijms-25-00933-f004:**
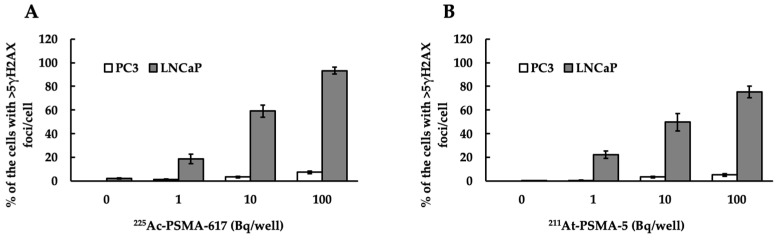
Percentage of cells with >5γH2AX foci/cells. (**A**) The white bar represents PC3 (PSMA^low^) cells treated with ^225^Ac-PSMA-617, the gray bar represents LNCaP (PSMA^high^) cells treated with ^225^Ac-PSMA-617. (**B**) the white bar represents PC3 cells treated with ^211^At-PSMA-5, and the gray bar represents LNCaP cells treated with ^211^At-PSMA-5. The mean ± S.E of the triplicate results is shown.

**Figure 5 ijms-25-00933-f005:**
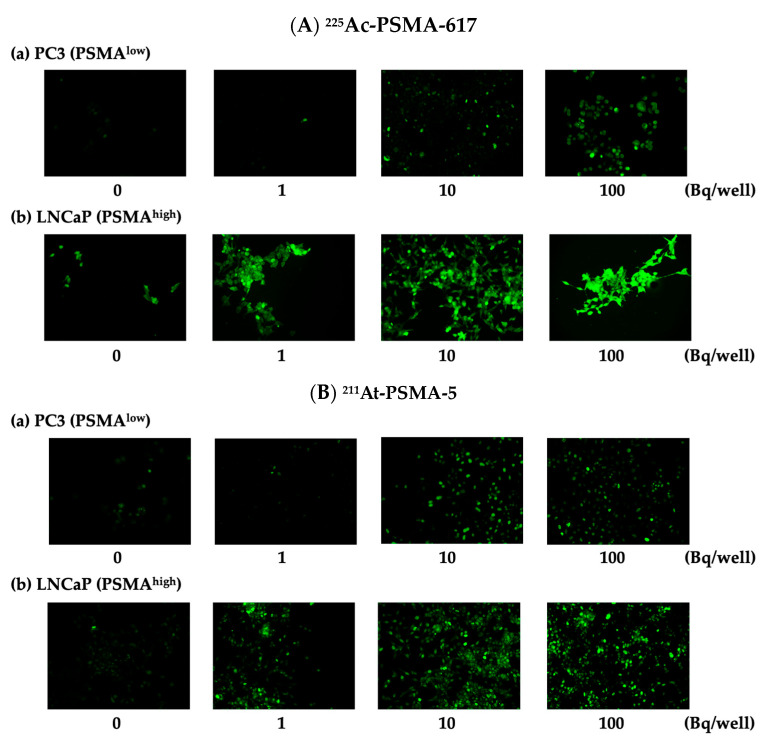
DSB induction of the cell lines. (**A**) (**a**) PC3 (PSMA^low^) and (**b**) LNCaP (PSMA^high^) cells treated with ^225^Ac-PSMA-617. (**B**) (**a**) PC3 and (**b**) LNCaP cells treated with ^211^At-PSMA-5.

**Figure 6 ijms-25-00933-f006:**
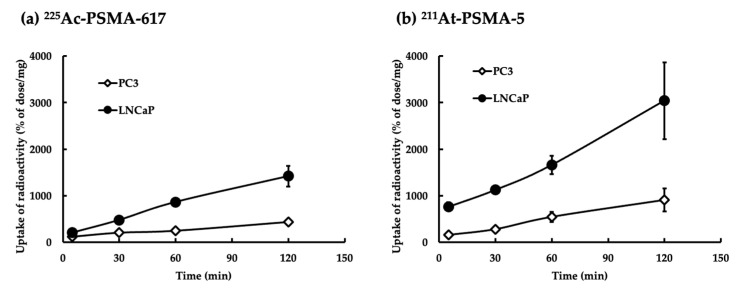
Uptake of the cell lines. (**a**) PC3 (PSMA^low^) and LNCaP (PSMA^high^) cells treated with ^225^Ac-PSMA-617. (**b**) PC3 and LNCaP cells treated with ^211^At-PSMA-5. The y axis indicates the uptake of radioactivity (% of dose/mg protein). The mean ± S.E of the triplicate results is shown.

**Figure 7 ijms-25-00933-f007:**
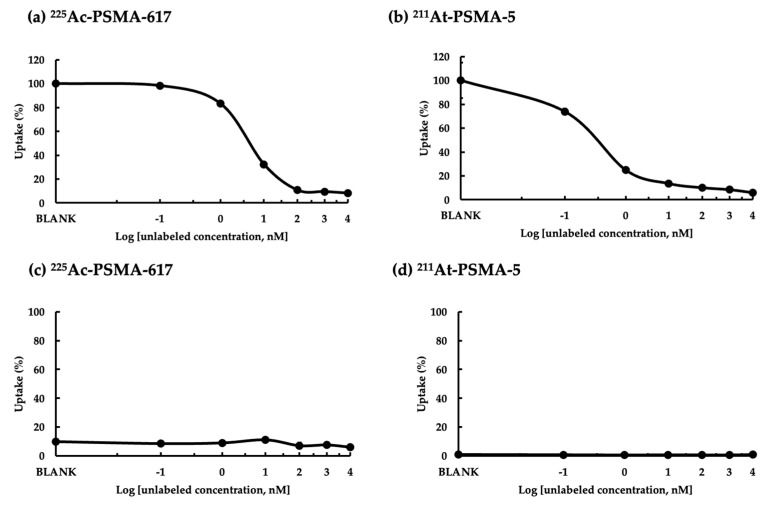
Inhibition of uptake of labeled PSMA in LNCaP (PSMA^hgh^) cells. Inhibition of (**a**) ^225^Ac-PSMA-617 and (**b**) ^211^At-PSMA-5 with unlabeled chemicals. The same experiment was performed in PC3 cells (PSMA^low^). Inhibition of (**c**) ^225^Ac-PSMA-617 and (**d**) ^211^At-PSMA-5 with unlabeled chemicals.

**Figure 8 ijms-25-00933-f008:**
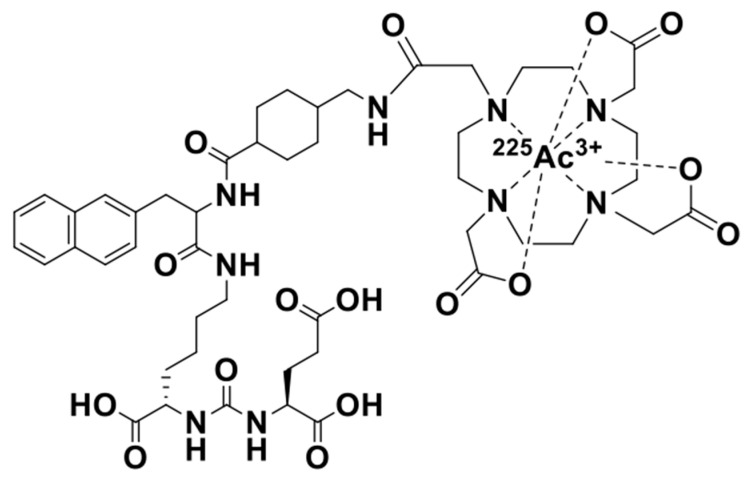
Structure of ^225^Ac-PSMA-617 radiotherapeutics.

**Figure 9 ijms-25-00933-f009:**
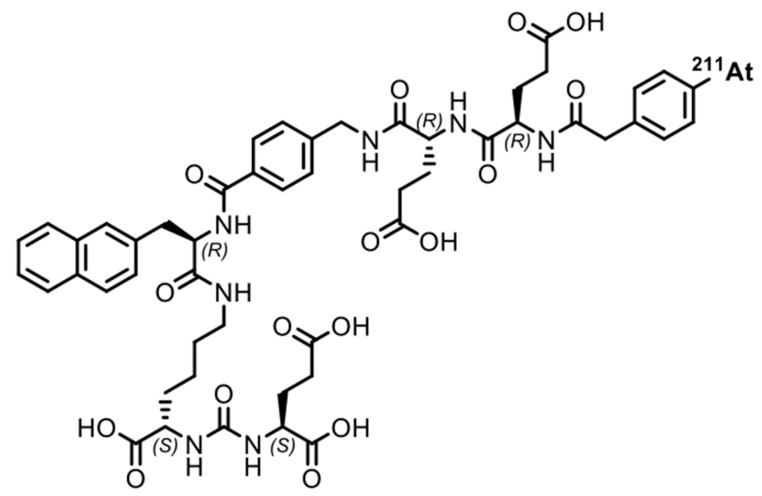
Structure of ^211^At-PSMA5 radiotherapeutics.

**Table 1 ijms-25-00933-t001:** TAT Radionuclides.

Nuclide	Half-Life	Decay	Stable
^227^Th/^223^Ra	18.7 days/11.4 days	α, β^−^	^207^Pb
^225^Ac/^213^Bi	10 days/45.6 min	α, β^−^	^209^Bi
^211^At	7.2 h	α, EC	^207^Pb
^212^Pb/^212^Bi	10.6 h/60.6 min	α, β^−^	^208^Pb
^230^U/^226^Th	20.8 days/30.6 min	α, β^−^, EC	^206^Pb
^149^Tb	4.1 h	α, EC	^145^Nd

Th: thorium, Ra: radium, Tb: terbium, Nd: neodymium, Ac: actinium, Bi: bismuth, At: astatine, Pb: lead, U: uran.

## Data Availability

Raw data were generated at Osaka University. Derived data supporting the findings of this study are available from the corresponding author [K.K.-N.] on request.

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
