# Peer review of "Comparison of Nuclear Medicine Therapeutics Targeting PSMA among Alpha-Emitting Nuclides"

_ijms, 2024, doi:10.3390/ijms25020933_

Round 1
Reviewer 1 Report
Comments and Suggestions for Authors
The authors present a thorough comparison between 225Ac-PSMA-617 and 211At-PSMA-5, highlighting their cytotoxic effects, DNA damage induction, etc. The research holds promise; however, I believe several key improvements would significantly enhance its value. I encourage the authors to consider the following suggestions:
- Ac-225 and At-211 are not the main alpha emitters. Tb-149, Bi-212/213, Pb-212, Ra-223 among others, also exist. I understand that these might not be mentioned in the abstract. Therefore, I would suggest including them in the introduction and writing a brief introductory sentence about targeted alpha therapy. This should be supported by relevant and up-to-date literature citations.
- I also recommend incorporating a more comprehensive introduction that contextualizes the research within the current landscape of prostate cancer treatment. A brief overview discussing existing therapies, their limitations, and the significance of PSMA as a therapeutic target would greatly benefit readers, providing a stronger foundation for the subsequent discussion.
- While the introduction touches upon the differences between 225Ac and 211At, structuring a more detailed comparison and advantages as isotopes would significantly bolster the manuscript's introductory section.
- Change "labeled chemicals" throughout the manuscript. Providing the specific names of these compounds would ensure clarity.
- The discussion on differences in uptake mechanisms based on chemical structure variations would be strengthened by suggesting additional experiments or hypotheses elucidating specific mechanisms behind these disparities in cellular uptake.
- Although the cytotoxic levels of the two nuclides and the radiation-induced DNA damage are highlighted, there is a lack of in-depth correlation with the clinical implications of these findings. A clear connection between the experimental data and the practical application of these nuclides in nuclear medicine is missing.
Author Response
EDITOR’S SPECIFIC COMMENTS
Reviewer 1
[Comment-1]
-The authors present a thorough comparison between 225Ac-PSMA-617 and 211At-PSMA-5, highlighting their cytotoxic effects, DNA damage induction, etc. The research holds promise; however, I believe several key improvements would significantly enhance its value. I encourage the authors to consider the following suggestions:
[Response-1]
Thank you very much for your kindly advises. Based on your kind advice, we had modified our article as follows. In addition, any additions or corrections made to the text compared to the first edition are highlighted in the manuscript.
[Comment-2]
-Ac-225 and At-211 are not the main alpha emitters. Tb-149, Bi-212/213, Pb-212, Ra-223 among others, also exist. I understand that these might not be mentioned in the abstract. Therefore, I would suggest including them in the introduction and writing a brief introductory sentence about targeted alpha therapy. This should be supported by relevant and up-to-date literature citations.
[Response-2]
Thank you for your kind comments.
We rewrote the abstract and made additions to the introduction following your advice.
[Comment-3]
- I also recommend incorporating a more comprehensive introduction that contextualizes the research within the current landscape of prostate cancer treatment. A brief overview discussing existing therapies, their limitations, and the significance of PSMA as a therapeutic target would greatly benefit readers, providing a stronger foundation for the subsequent discussion.
[Response-3]
Thank you very much for your advice. Prostate‐specific membrane antigen (PSMA) is highly expressed in poorly differentiated, metastatic, and castration‐resistant prostate cancers. While it is attracting attention as a therapeutic and diagnostic target, the pathophysiological functions of PSMA in prostate tumors remain unclear. We add the description about PSMA in introduction.
[Comment-4]
- While the introduction touches upon the differences between 225Ac and 211At, structuring a more detailed comparison and advantages as isotopes would significantly bolster the manuscript's introductory section.
[Response-4]
Thank you for your helpful suggestions. We showed these differences in Table 2 in introduction.
[Comment-5]
- Change "labeled chemicals" throughout the manuscript. Providing the specific names of these compounds would ensure clarity.
[Response-5]
Thank you very much. We rewrote them.
[Comment-6]
- The discussion on differences in uptake mechanisms based on chemical structure variations would be strengthened by suggesting additional experiments or hypotheses elucidating specific mechanisms behind these disparities in cellular uptake.
[Response-6]
PSMA is a transmembrane glycoprotein, and its expression increases in proportion to the malignancy and stage of prostate cancer. Since PSMA has glutamic acid hydrolysis activity, preparations based on glutamic acid-containing asymmetric urea-type compounds (Glu-Urea-based PSMA ligands, like PSMA-617) that bind to PSMA have been developed. PSMA preparations can be divided into those in which metal nuclides (68Ga,177Lu, and 225Ac) are chelated and those in which radioactive isotopes are incorporated into the compound. The effectiveness as a nuclear medicine therapeutic drug is proportional to its internalization into cells, and the stability depends on the structure of the compound or the method of labeling the nuclide. Thus, it is very difficult to determine the appropriate compound.
The purpose of this research is to compare studies using different alpha-emitting nuclides for the same molecular target, and to show that both nuclides can be used in the same way, although there are differences depending on the element or compound. This is to demonstrate the possibility of using easy-to-use nuclides. We thought that it is not suitable that the relationship between structure and usefulness in this article.
[Comment-7]
- Although the cytotoxic levels of the two nuclides and the radiation-induced DNA damage are highlighted, there is a lack of in-depth correlation with the clinical implications of these findings. A clear connection between the experimental data and the practical application of these nuclides in nuclear medicine is missing.
[Response-7]
225Ac-PSMA-617 has already been administered to humans, but 211At-PSMA-5 has not been administered to humans yet. On the other hand, there have not been much research of animal experiments on 225Ac-PMSA-617, and our group has reported on animal experiments on 211At-PSMA-5. Therefore, the purpose of this study was to first perform a comparison in comparable cell experiments and see what kind of differences there are between these nuclides. Strictly speaking, the compounds themselves are also different. This was because the nature of the nuclide required the use of different labeling methods. Thus, the object of research is not to show the connection between the experimental data and the practical application of these nuclides in nuclear medicine. By showing that each 211At or 225Ac can be used equally, we believe that we have been able to show that it is possible to select a nuclide that is easy to introduce depending on the circumstances of the clinical site.
Reviewer 2 Report
Comments and Suggestions for Authors
This is an interesting publication comparing radioconjugates labeled with 225Ac and 211At. However, I have several comments on this work that should be taken into account before publication.
-
1. Fig. 1: A normal scale would be much more preferable than a logarithmic one. Additionally, representing the number of alpha decays on the x-axis would be more effective than depicting the activity.
-
2. It will be necessary to include a figure depicting specific and nonspecific binding. In the case of alpha cytotoxicity, nonspecific binding is also significant.
-
3. The experimental section lacks information on the production of 211At, the energy of alpha particles used for irradiation, target preparation, and methods for separating astatine.
-
4. There is insufficient information regarding the stability of the radioconjugates in human serum, as well as in the medium during cell studies
Author Response
EDITOR’S SPECIFIC COMMENTS
Reviewer-2
This is an interesting publication comparing radioconjugates labeled with 225Ac and 211At. However, I have several comments on this work that should be taken into account before publication.
Thank you very much for your kindly advises.
[Comment-1]
Fig. 1: A normal scale would be much more preferable than a logarithmic one. Additionally, representing the number of alpha decays on the x-axis would be more effective than depicting the activity.
[Response-2]
We changed the y-axis in Figure 1 from logarithmic scale to normal scale.
[Comment-2]
It will be necessary to include a figure depicting specific and nonspecific binding. In the case of alpha cytotoxicity, nonspecific binding is also significant.
[Response-2]
For evaluate nonspecific binding, we performed the inhibition assay using PC3. We added the data from the study using PC3 in Figure 7. Inhibition of concentration-dependent uptake by unlabeled substances could not be confirmed.
[Comment-3]
The experimental section lacks information on the production of 211At, the energy of alpha particles used for irradiation, target preparation, and methods for separating astatine.
[Response-3]
We had already reported the procedure of 211At production in previous reports. We added the description about 211At separation in materials and methods section.
[Comment-4]
There is insufficient information regarding the stability of the radioconjugates in human serum, as well as in the medium during cell studies
[Response-4]
The stability of 211At-PSMA5 was evaluated by HPLC, TLC, and electrophoresis. Stability in blood or urine, no metabolites or degradation products were detected in vitro experiments. In vivo experiments, slight (<1%) deastatination was observed. The data itself was not shown because it is used in another article being submitted, but we have confirmed that the compounds were stable.
Round 2
Reviewer 2 Report
Comments and Suggestions for Authors
The publication can be published in the present form.